# Simplified and Unified Analysis of Various Learning Problems by Reduction to Multiple-Instance Learning

**Daiki Suehiro**[1, 2]                    **Eiji Takimoto**[3]

[1]Kyushu University, Department of Advanced Information Technology, 744 Motooka, Fukuoka, Japan
[2]RIKEN, Center for Advanced Intelligence Project, Nihonbashi 1-chome Mitsui Building, 15th floor,1-4-1 Nihonbashi, Chuo-ku, Tokyo, Japan
[3]Kyushu University, Department of Informatics,744 Motooka, Fukuoka, Japan

## Abstract

In statistical learning, many problem formulations have been proposed so far, such as multi-class learning, complementarily labeled learning, multi-label learning, multi-task learning, which provide theoretical models for various real-world tasks. Although they have been extensively studied, the relationship among them has not been fully investigated. In this work, we focus on a particular problem formulation called Multiple-Instance Learning (MIL), and show that various learning problems including all the problems mentioned above with some of new problems can be reduced to MIL with theoretically guaranteed generalization bounds, where the reductions are established under a new reduction scheme we provide as a by-product. The results imply that the MIL-reduction gives a simplified and unified framework for designing and analyzing algorithms for various learning problems. Moreover, we show that the MIL-reduction framework can be kernelized.

## 1 INTRODUCTION

In this study, we explore how a large class of learning problems can be reduced to the Multiple-Instance Learning (MIL) problem. This is strongly motivated by the results of [Sabato and Tishby, 2012] and [Suehiro et al., 2020]. Suehiro et al. [2020] showed that some local-feature-based learning problems can be reduced to a MIL problem, which gave us an insight that MIL would have a high capability of representing various learning problems. Indeed, the reduced problem is too specific whereas Sabato and Tishby [2012] proposed a much more general formulation of MIL, and thus we believe that a wider class of learning problems can be reduced to MIL.

We provide a MIL-reduction scheme and reveal that various learning problems, such as multi-class learning, complementarily labeled learning, multi-label learning, and multi-task learning, can be reduced to MIL. By the reduction, we immediately derive generalization bounds from [Sabato and Tishby, 2012], as well as learning algorithms. That is, our reduction scheme greatly *simplifies* the analyses of generalization bounds as compared with the analyses in the previous works [e.g., Lei et al., 2019, Ishida et al., 2017, Yu et al., 2014, Pontil and Maurer, 2013]. Some of the obtained generalization bounds are competitive or incomparable to the existing results. In particular, for multi-label learning, we derive an improved generalization bound, and for complementarily labeled learning, we derive a novel learning algorithm, which is the first polynomial-time algorithm in a certain setting. Moreover, we propose three new learning problems, *multi-label learning with perfectionistic loss*, *top-1 ranking learning* and *top-1 ranking learning with negative feedback*, and we demonstrate that they can be reduced to MIL as well. The results imply that our MIL-reduction gives a *unified scheme* for designing and analyzing algorithms for various learning problems.

To provide the MIL-reduction scheme, we propose a general reduction scheme among learning problems. Our scheme has two remarkable features as described below. First, our reduction transforms every instance-label pair $(x, y)$ in the given sample of the original learning problem to an instance-label pair $(x', y')$ to form a sample of the reduced learning problem. In contrast, standard reduction schemes employ an instance transformation and an label transformation separately, to construct $x'$ from $x$ and $y'$ from $y$, respectively. Therefore, our scheme enables us to design reduction algorithms among a wider class of learning problems, e.g., learning-to-rank to classification, and supervised learning to weakly supervised learning. Second, our reduction scheme ensures that the Empirical Risk Minimization (ERM) of the reduced problem implies the ERM of the original one, while the empirical Rademacher complexity of the hypothesis (composed with loss function) classes are preserved through the reduction. This means that we can employ an

*Accepted for the 38th Conference on Uncertainty in Artificial Intelligence* (UAI 2022).

existing ERM algorithm for the reduced problem to obtain an ERM algorithm for the original problem with a theoretical guaranteed generalization bound, which is immediately derived from a known generalization bound for the reduced problem. We also show that the MIL-reduction scheme can be kernelized.

The main contributions are summarized as follows:

- We propose a general reduction scheme based on the ERM, which allows us to derive a generalization risk bound of the original problem immediately.
- We demonstrate that several learning problems, from traditional to new problems, can be reduced to MIL. The results imply that our MIL-reduction gives a simplified and unified scheme for the analyses for various learning problems.
- We obtain novel theoretical results for some learning problems.
- We show that the MIL-reduction scheme can be kernelized.

Several proofs are shown in supplementary materials.

## 2 PRELIMINARIES

For an integer $u$, $[u]$ denotes the set $\{1, \ldots, u\}$. $I(e)$ denotes the indicator function of the event e, that is, $I(e) = 1$ if e is true and $I(e) = 0$ otherwise.

A learning problem is represented by a pair $(\mathcal{H}, \ell)$ of a hypothesis class $\mathcal{H} \subseteq \{h : \mathcal{X} \to \mathcal{Y}\}$ and a loss function $\ell : \mathcal{X} \times \mathcal{Y} \times \mathcal{H} \to \mathbb{R}$ for some input space $\mathcal{X}$ and output space $\mathcal{Y}$. A learner receives a sample $S = ((x_1, y_1), \ldots, (x_n, y_n))$ where each input-output pair $(x_i, y_i)$ is drawn i.i.d. according to an unknown distribution $\mathcal{D}$ over $\mathcal{X} \times \mathcal{Y}$. The goal of the learner is to find, with high probability, a hypothesis $h \in \mathcal{H}$ so that the generalization risk $R_{\mathcal{D}}(h) = \mathbb{E}_{(x,y) \sim \mathcal{D}} \ell(x, y, h)$ is small. For a learning problem $(\mathcal{H}, \ell)$, we define a class of loss functions as $\widehat{\mathcal{H}} = \{(x, y) \mapsto \ell(x, y, h) \mid h \in \mathcal{H}\}$ when the underlying loss function $\ell$ is clear from the context. We give the definition of the empirical Rademacher complexity, which is used to bound the generalization risk.

**Definition 1** (Empirical Rademacher complexity [Bartlett and Mendelson, 2003]). *Given a sample* $S = ((x_1, y_1), \ldots, (x_n, y_n)) \in (\mathcal{X} \times \mathcal{Y})^n$, *the empirical Rademacher complexity* $\mathfrak{R}_S(\widehat{\mathcal{H}})$ *of a class* $\widehat{\mathcal{H}}$ *w.r.t. S is defined as* $\mathfrak{R}_S(\widehat{\mathcal{H}}) = \frac{1}{n}\mathbb{E}_{\boldsymbol{\sigma}}\left[\sup_{g \in \widehat{\mathcal{H}}} \sum_{i=1}^{n} \sigma_i g(x_i, y_i)\right]$, *where* $\boldsymbol{\sigma} \in \{-1, 1\}^n$ *and each* $\sigma_i$ *is an independent uniform random variable taking values in* $\{-1, +1\}$.

**Generalization risk bound [Mohri et al., 2018]** Let $(\mathcal{H}, \ell)$ be a learning problem and $S$ be a sample of size

$n$ drawn according to a distribution $\mathcal{D}$. Then, it holds with probability at least $1 - \delta$ that for all $h \in \mathcal{H}$,

$$R_{\mathcal{D}}(h) \leq \widehat{R}_S(h) + 2\mathfrak{R}_S(\widehat{\mathcal{H}}) + 3\sqrt{\log(2/\delta)/2n},$$

where $\widehat{R}_S(h) = \frac{1}{n}\sum_{i=1}^{n} \ell(x_i, y_i, h)$ denotes the empirical risk of $h$ for sample $S$.

## 3 REDUCTION SCHEME FOR ERM

We propose a general reduction scheme for empirical risk minimization and provide useful theoretical results.

**Definition 2** (ERM-reduction). *A learning problem* $(\mathcal{H}, \ell)$ *over input-output space* $\mathcal{X} \times \mathcal{Y}$ *is* ERM-reducible *to another learning problem* $(\mathcal{H}', \ell')$ *over input-output space* $\mathcal{X}' \times \mathcal{Y}'$ *if there exist polynomial-time computable functions* $\alpha : \mathcal{X} \times \mathcal{Y} \to \mathcal{X}' \times \mathcal{Y}'$ *and* $\beta : \mathcal{H}' \to \mathcal{H}$ *such that for any* $(x, y) \in \mathcal{X} \times \mathcal{Y}$ *and for any* $h' \in \mathcal{H}'$,

$$\ell(x, y, h) = \ell'(x', y', h'),$$

*where* $(x', y') = \alpha(x, y)$ *and* $h = \beta(h')$.

Here we show the remarkable relationship between the original problem and the reduced problem.

**Proposition 1.** *Suppose that* $(\mathcal{H}, \ell)$ *is ERM-reducible to* $(\mathcal{H}', \ell')$ *with transformations* $\alpha$ *and* $\beta$. *For any sample* $S = ((x_1, y_1), \ldots, (x_n, y_n)) \in (\mathcal{X} \times \mathcal{Y})^n$, *the following holds:*

*(i) (In)equality of the ERMs:*

$$\min_{h \in \mathcal{H}} \widehat{R}_S(h) \leq \min_{h \in \mathcal{H}_\beta} \widehat{R}_S(h)$$
$$= \min_{h' \in \mathcal{H}'} \widehat{R}_{S'}(h'),$$

*where* $\mathcal{H}_\beta = \{\beta(h') \mid h' \in \mathcal{H}'\}$ *and* $S' = ((x_1', y_1'), \ldots, (x_n', y_n'))$ *with* $(x_i', y_i') = \alpha(x_i, y_i)$ *for* $i \in [n]$.

*(ii) Empirical Rademacher complexity preserving:*

$$\mathfrak{R}_S(\widehat{\mathcal{H}}_\beta) = \mathfrak{R}_{S'}(\widehat{\mathcal{H}'}).$$

We can design a reduction scheme in a straightforward way as follows. When given a sample $S$ of the original problem, we construct $S'$ of the reduced problem by $\alpha$ and obtain $h'$ by solving the ERM of the reduced problem. Then, we obtain the final hypothesis $h$ by $\beta$.

We derive the following generalization risk bound using the propositions on the empirical Rademacher complexity.

**Corollary 2.** *Let* $S = ((x_1, y_1), \ldots, (x_n, y_n))$ *be a sample i.i.d. drawn according to unknown distribution* $\mathcal{D}$ *in an original problem* $(\mathcal{H}, \ell)$. *If* $(\mathcal{H}, \ell)$ *is ERM-reducible to* $(\mathcal{H}', \ell')$, *for* $S' = (\alpha(x_1, y_1), \ldots, \alpha(x_n, y_n))$ *and* $h = \beta(h')$, *the following generalization risk bound holds with a probability at least* $1 - \delta$ *for all* $h \in \mathcal{H}_\beta$:

$$R_{\mathcal{D}}(h) \leq \widehat{R}_{S'}(h') + 2\mathfrak{R}_{S'}(\widehat{\mathcal{H}'}) + 3\sqrt{\log(2/\delta)/2n}.$$

That is, we can guarantee the generalization bound of the original problem because of the preservation of the empirical Rademacher complexity.

# 4 MIL-REDUCTION FRAMEWORK

This section is the highlight of this paper. We define the ERM-reducibility to MIL and show the reducible condition. Moreover, we show that some theoretical analyses can be simplified. We use some symbols with prime (e.g., $\mathcal{X}'$) to indicate that the MIL is the reduced problem.

## 4.1 PROBLEM FORMULATION OF MIL

Let $\mathcal{Z} \subseteq \mathbb{R}^{d'}$ be the instance space. $\mathcal{X}' \subseteq 2^{\mathcal{Z}}$ is an input space and a *bag* $x' \in \mathcal{X}'$ is a finite set of instances chosen from $\mathcal{Z}$. Let $\mathcal{Y}' = \{-1, 1\}$ be an output space. Following the formulation by [Sabato and Tishby, 2012], we define, for the rest of the paper, a MIL problem as a pair $(\mathcal{H}', \ell')$ of a hypothesis class $\mathcal{H}'$ and a loss function $\ell'$ of the form:

$$\mathcal{H}' = \{h' : x' \mapsto \Psi_p(\{f_2(g(z)) \mid z \in x'\}) \mid g \in \mathcal{G}\}, \quad (1)$$

$$\ell' : (x', y', h') \mapsto f_1(y'h'(x')), \quad (2)$$

where $\mathcal{G} \subseteq \{g : \mathcal{Z} \to \mathbb{R}\}$, $f_1 : \mathbb{R} \to [0, 1]$ is an $a$-Lipschitz function, $f_2 : \mathbb{R} \to [-1, 1]$ is a $b$-Lipschitz function, and $\Psi_p : 2^{[-1,1]} \to [-1, 1]$ is a $p$-norm like function, which is defined for any $p \in [1, \infty)$ as

$$\Psi_p(V) = \left(\frac{1}{m} \sum_{i=1}^{m} (v_i + 1)^p\right)^{1/p} - 1$$

for every finite set $V = \{v_1, v_2, \ldots, v_m\} \subseteq [-1, 1]$. We define $\Psi_\infty$ as $\lim_{p \to \infty} \Psi_p$. Note that $\Psi_p$ is 1-Lipschitz for any $p$ [see, Sabato and Tishby, 2012]. In MIL tasks, $\Psi_p$ is a user-defined function and behaves as an aggregation of some bag information. Typical $\Psi_p$ are the max operator ($p = \infty$) and average ($p = 1$).

The only difference in the hypothesis of [Sabato and Tishby, 2012] is $f_2$. $f_2$ appears redundant (because $f_2 \circ g$ can be replaced by a single function) but plays an important role in the reduction (the examples are shown in Section 5).

Here we give the definition of ERM-reducibility in a straightforward way.

**Definition 3** (MIL-reducibility). *A learning problem $(\mathcal{H}, \ell)$ is said to be MIL-reducible if there exists a MIL problem $(\mathcal{H}', \ell')$ such that $(\mathcal{H}, \ell)$ is ERM-reducible to $(\mathcal{H}', \ell')$.*

Hereinafter, the scheme for ERM-reduction to MIL is called *MIL-reduction scheme*.

## 4.2 RADEMACHER COMPLEXITY BOUND

We show the empirical Rademacher complexity bound for the MIL-reducible problems using our reduction scheme. As aforementioned, the main advantage of our reduction scheme is to allow us to apply the empirical Rademacher complexity bound of the reduced problem to the original problems. In this paper, we utilize the bound provided by Sabato and Tishby [2012].

**Theorem 3** (An application of Theorem 20 of [Sabato and Tishby, 2012]). *Let $(\mathcal{H}', \ell')$ be a MIL problem defined in Eq.(1) and (2). Let $S' = ((x'_1, y'_1), \ldots, (x'_n, y'_n))$ be a sample with average bag size $r_{S'}$. Let $\widehat{\mathcal{G}} = \{f_2 \circ g \mid g \in \mathcal{G}\}$. If there exist $C, \rho \geq 0$ such that for all sufficiently large $n$,*

$$\mathfrak{R}_{S'}(\widehat{\mathcal{G}}) \leq \frac{C \ln^\rho(n)}{\sqrt{n}},$$

*then*

$$\mathfrak{R}_{S'}(\widehat{\mathcal{H}'}) = O\left(\frac{\log\left(a^2 n^2 r_{S'}\right)\left(\frac{aC}{\rho+1} \ln^{\rho+1}(a^2 n)\right)}{\sqrt{n}}\right),$$

*where $\widehat{\mathcal{H}'} = \{\hat{h}' : x' \mapsto f_1(y'h'(x')) \mid h' \in \mathcal{H}'\}$.*

As mentioned in [Sabato and Tishby, 2012], we obtain the following bound when $\mathcal{G}$ is a set of linear functions.

**Corollary 4.** *Let $\mathcal{G} = \{g : z \mapsto \langle w', z \rangle \mid w' \in \mathbb{R}^{d'}, \|w'\| \leq C_1\}$ and assume that $\|z\| \leq C_2$. Then, the following bound holds:*

$$\mathfrak{R}_{S'}(\widehat{\mathcal{H}}) = O\left(\frac{\log\left(a^2 n^2 r_{S'}\right)\left(ab C_1 C_2 \ln(a^2 n)\right)}{\sqrt{n}}\right).$$

The above bound is easily derived from the result of $\mathfrak{R}_{S'}$ [see the proof of Theorem 20 of Sabato and Tishby, 2012]) and $\mathfrak{R}_{S'}(\widehat{\mathcal{G}}) \leq b\mathfrak{R}_{S'}(\mathcal{G}) \leq {}^{bC_1 C_2}/\sqrt{n} = {}^{bC_1 C_2 \ln^0(n)}/\sqrt{n}$ [see, e.g., Theorem 5.8 and 5.10 of Mohri et al., 2018].

Using Theorem 3 and Corollary 2, we obtain a generalization risk bound for MIL-reducible problems.

## 4.3 LEARNING ALGORITHM

We show that, under mild conditions, the ERM of MIL becomes a convex or a DC (Difference of Convex) programming problem. Suppose that $\mathcal{G}$ is a set of linear functions:

$$\mathcal{G} = \{g : z \mapsto \langle w', z \rangle \mid w' \in \mathbb{R}^{d'}, \|w'\| \leq C_1\}. \quad (3)$$

Let $S' = ((x'_1, y'_1), \ldots, (x'_n, y'_n))$. The ERM of MIL is formulated as follows:

$$\min_{\|w'\| \leq C_1} \lambda \|w'\|^2 + \sum_{i=1}^{n} f_1\left(y'_i \Psi_p\left(\{f_2\left(\langle w', z \rangle \mid z \in x'_i\right)\}\right)\right). \quad (4)$$

For the optimization problem (4), we show that the following propositions hold.

**Proposition 5.** *If $y_i' = -1$ for any $i \in [n]$ for sample $S'$, $f_1$ is convex and nonincreasing [1], and $f_2$ is a nondecreasing convex function, and $\mathcal{G}$ is given as Eq.(3), then the ERM of $(\mathcal{H}', \ell')$ is a convex programming problem.*

**Proposition 6.** *If $f_1$ is a nonincreasing convex [1] and $f_1(c)$ is a homogeneous function of degree 1 for $c \in [-1, 1]^2$, and $f_2$ is a nondecreasing convex function, and $\mathcal{G}$ is given as Eq.(3), then ERM of $(\mathcal{H}', \ell')$ is a DC programming problem.*

Generally, it is difficult to find a global minimum for a DC programming problem; however, it is known that we can find a solution with $\epsilon$-approximation of local optima [see, e.g., Le Thi and Dinh, 2018]. We introduce a standard DC algorithm to solve (4) in Algorithm 1 in Sec. D.

The propositions indicate that, if $(\mathcal{H}, \ell)$ is MIL-reducible to $(\mathcal{H}', \ell')$ and satisfies either of the above conditions, then the solution $h \in \mathcal{H}_\beta$ in the original problem can be obtained by a unified learning algorithm.

# 5 MIL-REDUCIBLE EXAMPLES

In this section, we demonstrate that various learning problems can be reduced to MIL by the proposed reduction scheme. The results imply that our MIL-reduction gives a unified scheme for designing and analyzing learning algorithms for various learning problems [3].

## 5.1 THE EXISTING PROBLEMS

### 5.1.1 Multi-class learning problem

**Problem setting:** Let $\mathcal{X} \subseteq \mathbb{R}^d$ be an instance space, and $\mathcal{Y} = [k]$ be an output space. The learner receives the set of labeled instances $S = ((x_1, y_1), \ldots, (x_n, y_n)) \in (\mathcal{X} \times \mathcal{Y})^n$, where each instance is drawn i.i.d. according to some unknown distribution $\mathcal{D}$. The learner predicts the label of $x$ using the hypothesis $h \in \mathcal{H} = \{x \mapsto \arg\max_{j \in [k]} \langle w_j, x \rangle \mid \forall j \in [k], w_j \in \mathbb{R}^d\}$. Let $\ell : (x, y, h) \mapsto \Gamma(\langle w_y, x \rangle - \max_{j \in \mathcal{Y} \backslash y} \langle w_j, x \rangle)$ be a loss function, where $\Gamma : \mathbb{R} \to [0, 1]$ is a convex, nonincreasing and $a$-Lipschitz function. The generalization risk and empirical risk of $h$ are defined as:

$$R_\mathcal{D}(h) = \mathop{\mathbb{E}}_{(x,y) \sim \mathcal{D}} \ell(x, y, h), \widehat{R}_S(h) = \frac{1}{n} \sum_{i=1}^{n} \ell(x_i, y_i, h).$$

---

[1]More precisely, the extended-value extension $f_1$ also must be nonincreasing (See details in [Boyd and Vandenberghe, 2004]).

[2]For example, hinge-loss function $f(c) = \max\{0, 1 - c\}$ satisfies this condition.

[3]The reduction of multi-task learning and top-1 ranking learning negative feedback are shown in Sec.G and J owing to space limitations.

We obtain the following by using MIL-reduction scheme:

**Theorem 7.** *Multi-class learning problem is MIL-reducible.*

*Proof.* For any $(x, y)$, we define

$$\eta_{(x,y)} = (\mathbf{0}, \ldots, \mathbf{0}, \underbrace{x}_{y-\text{th block}}, \mathbf{0}, \ldots, \mathbf{0}),$$

where $\mathbf{0}$ is a $d$-dimensional vector, the elements of which are all 0. On the MIL-reduction scheme, suppose that $p = \infty$; $f_1(c) = \Gamma(2cC_1C_2)$, $f_2(c) = c/2C_1C_2$ (shifting function to $[-1, +1]$); $\alpha(x, y) = (x'_{(x,y)}, y')$ where $x'_{(x,y)} = \{\eta_{(x,j)} - \eta_{(x,y)} \mid \forall j \in \mathcal{Y} \backslash y\}$; $y' = -1$; for any $z \in \mathbb{R}^{kd}$, $\mathcal{G} = \{g : z \mapsto \langle (w_1', \ldots, w_k'), z \rangle \mid w_j' \in \mathbb{R}^d, \forall j \in [k], \|W'\| \leq C_1\}$ where $W' = (w_1', \ldots, w_k')$ and $\|W'\| = \sqrt{\sum_{j=1}^{k} \|w_j'\|^2}$; $\beta(h') : x \mapsto \arg\max_{j \in [k]} \langle w_j', x \rangle$. Then, for any $(x, y)$ and $h \in \mathcal{H}$,

$$\ell'(x', y', h') = f_1\left(y' \Psi_p\left(\left\{f_2\left(g(z) \mid z \in x'_{(x,y)}\right)\right\}\right)\right)$$

$$= \Gamma\left(-\frac{1}{2C_1C_2} \Psi_\infty\left(\left\{2C_1C_2\left(g(z) \mid z \in x'_{(x,y)}\right)\right\}\right)\right)$$

$$= \Gamma\left(-\frac{1}{2C_1C_2} \max\left(2C_1C_2\left\{\left(g(z) \mid z \in x'_{(x,y)}\right)\right\}\right)\right)$$

$$= \Gamma\left(-\frac{2C_1C_2}{2C_1C_2} \max\left(\left\{\left(g(z) \mid z \in x'_{(x,y)}\right)\right\}\right)\right)$$

$$= \Gamma\left(-\left(\max\left\{\langle w', \eta_{(x,j)} - \eta_{(x,y)} \rangle \mid \forall j \in \mathcal{Y} \backslash y\right\}\right)\right)$$

$$= \Gamma\left(-\left(\max_{j \in \mathcal{Y} \backslash y}\left(\langle w_j, x \rangle - \langle w_y, x \rangle\right)\right)\right)$$

$$= \ell(x, y, h)$$

$\square$

The empirical Rademacher complexity is immediately derived as follows by observing the reduction process.

**Corollary 8.** *We assume that $\|x_i\| \leq C_2$ for any $i \in [n]$. In the reduced MIL problem from multi-class learning problem, the empirical Rademacher complexity of $\widehat{\mathcal{H}}'$ is given as:*

$$\mathfrak{R}_{S'}(\widehat{\mathcal{H}}') = O\left(\frac{\log\left(\hat{a}^2 2n^2(k-1)\right)\left(2\hat{a}\ln(\hat{a}^2 n)\right)}{\sqrt{n}}\right),$$

*where $\hat{a} = 2aC_1C_2$ and we assume $\|w'\| \leq C_1$ in the reduced MIL.*

We used the fact that the bag size is $(k - 1)$ for all $x_i'$ (i.e., $r_{S'} = k - 1$) and $\mathfrak{R}(\widehat{\mathcal{G}}) \leq 2/\sqrt{n}$ by setting $f_2(c) = c/C_1C_2$. Using Corollary 2, we can obtain the generalization risk bound for the multi-class learning.

The learning algorithm is obtained by the following result.

**Corollary 9.** *The reduced ERM of the MIL from multi-class learning is a convex programming problem.*

The proof of Theorem 7 shows that $f_2$ is nondecreasing convex and $y_i' = -1$ for all $i \in [n]$. Therefore, by Proposition 5, if we consider $\Gamma$ that is a nonicreasing and convex function, the ERM of the reduced MIL problem is a convex programming problem and solved in polynomial time.

### 5.1.2 Complementarily labeled learning problem

Complementarily labeled learning was proposed by Ishida et al. [2017]. In this problem, some training instances are complementarily labeled (e.g., instance $x_i$ is NOT $y_i$). We essentially follow the problem setting and some assumptions provided by Ishida et al. [2017].

**Problem setting:** Let $\mathcal{X} \subseteq \mathbb{R}^d$ be an instance space and $\mathcal{Y} = [k]$ be an output space. Let $\mathcal{D}$ be an unknown distribution over $\mathcal{X} \times \mathcal{Y}$. We assume that the learner receives a sample $S$ drawn i.i.d. according to the distribution $\mathcal{D}'$ which provides the true label with unknown probability $\theta$ and the complementary label with unknown probability $1 - \theta$. Moreover, we assume that the complementary label is chosen with a uniform probability (i.e., all complementary labels are equally chosen with the probability $1/(k-1)$). [4] More formally, we assume that the sample is given as $S = ((x_1, y_1, \gamma_1) \dots, (x_n, y_n, \gamma_n))$ which is drawn i.i.d. according to the distribution $\mathcal{D}'$ over $\mathcal{D} \times \{\text{False}, \text{True}\}$, where $\gamma_i = \text{True}$ means that $y_i$ is the true label and $\gamma_i = \text{False}$ means that $y_i$ is the complementary label (i.e., it indicates that $x_i$ is NOT $y_i$). For any $(x, y) \sim \mathcal{D}$, $\mathcal{D}'(x, y, \text{True}) = \theta$ and $\mathcal{D}'(x, \bar{y}, \text{False}) = \frac{1-\theta}{k-1}$ for any $\bar{y} \neq y$ (i.e., the complementary label is chosen with a uniform probability). The other basic settings are the same as those for the aforementioned multi-class learning. The learner predicts the label of $x$ using the hypothesis $h \in \mathcal{H} = \{x \mapsto \arg\max_{j \in [k]} \langle w_j, x \rangle \mid \forall j \in [k], w_j \in \mathbb{R}^d\}$. The final goal of the learner is to find $h \in \mathcal{H}$ with a small multi-class classification risk:

$$R_{\mathcal{D}}^{\text{MC}}(h) = \mathop{\mathbb{E}}_{(x,y)\sim\mathcal{D}} I\left(y \neq h(x)\right).$$

However, it is difficult to minimize the empirical multi-class classification risk directly using the complementarily labeled data. Therefore, we consider the following risk[5].

$$R_{\mathcal{D}'}^{\text{LC}}(h) = \mathop{\mathbb{E}}_{(x,y,\gamma)\sim\mathcal{D}'} \left[I\left(\gamma = (y \neq h(x))\right)\right].$$

This risk implies that when $\gamma = \text{True}$, the learner does not incur a risk if it predicts the true label. When $\gamma =$

---

[4]This assumption was proposed by Ishida et al. [2017] as a reasonable scenario in some practical tasks (e.g., crowdsourcing).

[5]Ishida et al. [2017] used a different surrogate risk. However, they and we have a common goal: to minimize $R_{\mathcal{D}}^{\text{MC}}(h)$.

False, the learner does not incur a risk if it predicts an assigned nontrue label. Thus, the risk measure is defined using the pair $(y, \gamma) \in (\mathcal{Y} \times \{\text{False}, \text{True}\})$. We can show that achieving a small $R_{\mathcal{D}'}^{\text{LC}}(h)$ is consistent with achieving small $R_{\mathcal{D}}^{\text{MC}}(h)$ as follows:

**Lemma 1.** *For any $h \in \mathcal{H}$, $R_{\mathcal{D}}^{\text{MC}}(h) = \frac{k-1}{\theta(k-2)+1} R_{\mathcal{D}'}^{\text{LC}}(h)$ holds.*

Thus, minimizing $R_{\mathcal{D}'}^{\text{LC}}(h)$ is a reasonable way to achieve a high multi-class classification accuracy.

Generally, there is no loss function $\ell((x, \gamma), y, h)$ which is a convex upper bound on the zero-one loss $I(\gamma = (y \neq h(x)))$ over the domain $w$. This is because if $I(\gamma = \text{True}) = 1$ then $\max$ is convex w.r.t. $w$; however, if $I(\gamma = \text{True}) = -1$ then $-\max = \min$ is concave w.r.t. $w$. Therefore, we consider the convex upper bounded loss only on the risk for complementarily labeled data (i.e., the concave risk for the normally labeled data) using $\Gamma : \mathbb{R} \to [0, 1]$ as $\Gamma\left(\max_{j \in \mathcal{Y}\setminus y}\langle(w_j - w_y), x\rangle\right)$. We then define the nonconvex risk $\ell(x, (\gamma, y), h) = \Gamma\left(I(\gamma = \text{True}) \times \left(\max_{j \in \mathcal{Y}\setminus y}\langle(w_j - w_y), x\rangle\right)\right)$. The empirical risk is formulated as:

$$\widehat{R}_S^{\text{LC}}(h) = \frac{1}{n}\sum_{i=1}^{n} \ell\left(x_i, (\gamma_i, y_i), h\right).$$

The following is obtained by MIL-reduction scheme.

**Theorem 10.** *Complementarily labeled learning is MIL-reducible.*

The difference from the reduction in multi-class learning is that only $y'$ takes $\{-1, 1\}$. $y'$ behaves as a *switch* that turns the loss of complementarily or normally labeled data.

The empirical Rademacher complexity is bounded as:

**Corollary 11.** *We assume that $\|x_i\| \leq C_2$ for any $i \in [n]$. In the reduced MIL problem from complementarily labeled learning, the empirical Rademacher complexity of $\widehat{\mathcal{H}}'$ is given by:*

$$\mathfrak{R}_{S'}(\widehat{\mathcal{H}}') = O\left(\frac{\log\left(\hat{a}^2 n^2(k-1)\right)\left(2\hat{a}\ln(\hat{a}^2 n)\right)}{\sqrt{n}}\right),$$

*where $\hat{a} = 2aC_1C_2$ and we assume $\|w'\| \leq C_1$ in the reduced MIL problem.*

We use the same argument as in Corollary 8. Using Corollary 2 and Lemma 1, we obtain the generalization bound for the complementarily labeled learning.

The learning algorithm is derived by the following result:

**Corollary 12.** *The reduced ERM of the MIL from complementarily labeled learning is a DC programming problem. If the sample contains only complementarily labeled data, the learning problem is a convex programming problem.*

Generally, $y' \in \{-1, 1\}$ in complementarily labeled learning. Using the proof of Theorem 10 and by Proposition 6, if we consider $\Gamma(c)$ which is a nondecreasing and homogeneous function of degree 1 for $c \in [-1, 1]$ such as hinge-loss function, we can solve the problem by DC algorithm as shown in Algorithm 1. Note that, if the sample contains only complementarily labeled data (i.e., $\forall i \in [n], y_i = -1$), it becomes a convex programming problem.

### 5.1.3 Multi-label learning problem

**Problem setting** Let $\mathcal{X} \subseteq \mathbb{R}^d$ be an instance space and $\mathcal{Y} \in \{-1, 1\}^k$ be an output space, and $\mathcal{D}$ be an unknown distribution over $\mathcal{X}$. Unlike the standard multi-class learning setting introduced in Section 5.1.1, each instance may have multiple labels (e.g., in text-categorization tasks, some texts have multiple topics such as IT and business). $y^j$ denotes the $j$-th element of $y_i$. The learner receives a labeled sample $S = (x_1, y_1), \ldots, (x_n, y_n) \in \mathcal{X} \times \mathcal{Y}$ which is drawn i.i.d. according to the distribution $\mathcal{D}$. The learner predicts whether $x$ belongs to class $j \in [k]$ or not using the hypothesis $h \in \mathcal{H} = \{(x, j) \mapsto \text{sign}(\langle w_j, x \rangle) \mid \forall w_j \in \mathbb{R}^d\}$. Let $\ell : (x, y, h) \mapsto \frac{1}{k} \sum_{j=1}^{k} \Gamma(-y^j \langle w_j, x \rangle)$ where $\Gamma : \mathbb{R} \to [0, 1]$ is a convex, nondecreasing and $b$-Lipschitz function [6]. The generalization and empirical risk of $h$ are defined as:

$$R_{\mathcal{D}}(h) = \mathop{\mathbb{E}}_{(x,y) \sim \mathcal{D}} [\ell(x,y,h)], \quad \widehat{R}_S(h) = \frac{1}{n} \sum_{i=1}^{n} \ell(x_i, y_i, h).$$

**Reduction to MIL**

**Theorem 13.** *Multi-label learning is MIL-reducible.*

*Proof.* On the MIL-reduction scheme, suppose that $p = 1$; $f_1 : f_1(a) = -a$ for $a \in \mathbb{R}$; $f_2$ is $\Gamma$; $\alpha(x, y) = (x'_{(x,y)}, y')$ where $x'_{(x,y)} = \{(-y^1 x, 1), \ldots, (-y^k x, k)\}$; $y' = -1$; $\mathcal{G} = \{g : (z, j) \mapsto \langle w'_j, z \rangle \mid w'_j \in \mathbb{R}^d, \forall j \in [k], \|W'\| \leq C_1\}$ where $W' = (w'_1, \ldots, w'_k)$; $\beta(h') : (x, j) \mapsto \text{sign}(\langle w'_j, x \rangle)$. For any $(x, y)$ and $h \in \mathcal{H}$, we have that

$$\ell'(x', y', h') = f_1 \left( y' \Psi_p \left( \left\{ f_2(g(z)) \mid z \in x'_{(x,y)} \right\} \right) \right)$$
$$= \frac{1}{|x'_{(x,y)}|} \sum_{(y^j x, j) \in x'_{(x,y)}} \Gamma \left( -\langle w_j, y^j x \rangle \right)$$
$$= \ell(x, y, h)$$

$\square$

The empirical Rademacher complexity is bounded as:

---

[6]Note that we use the negative score $-y^j \langle w_j, x \rangle$ to employ a nondecreasing $\Gamma$.

**Corollary 14.** *We assume that $\|x_i\| \leq C_2$ for any $i \in [n]$. In the reduced MIL problem, the empirical Rademacher complexity of $\widehat{\mathcal{H}}'$ is given as follows:*

$$\mathfrak{R}_{S'}(\widehat{\mathcal{H}}') = O\left( \frac{\log\left(2n^2 k\right)(bC_1 C_2 \ln(n))}{\sqrt{n}} \right),$$

*where $\|w'\| \leq C_1$ in the reduced MIL.*

We used the fact that the size of each bag is $k$. Using Corollary 2, we obtain the generalization risk bound for the multi-label learning.

The learning algorithm is obtained by the following result.

**Corollary 15.** *The reduced ERM of the MIL from multi-label learning is a convex programming problem.*

The proof of Theorem 13 shows that, $f_1$ is nonincreasing and convex, and $y'_i = -1$ for all $i \in [n]$. Therefore, by Proposition 5, if we consider $\Gamma$ that is nondecreasing and convex, the reduced problem is a convex programming problem and it is solved in polynomial time.

## 5.2 APPLICATION TO THE NEW PROBLEMS

### 5.2.1 Multi-label learning with perfectionistic loss

**Problem setting:** In a standard multi-label learning (see Sec.5.1.3), we consider the average prediction error (loss) with the classes. On the other hand, we consider a *perfectionistic* error in multi-label learning problem. More formally, we consider the following loss in a multi-label learning:

$$\ell : (x, y, h) \mapsto \max_{j \in [k]} \Gamma(-y^j \langle w_j, x \rangle),$$

where $\Gamma : \mathbb{R} \to [0, 1]$ is a convex, nondecreasing and $b$-Lipschitz function. This loss means that the learner incurs the risk unless the learner perfectly predict the correct labels. The generalization and empirical risks of $h$ are given as $R_{\mathcal{D}}(h) = \mathbb{E}_{(x,y) \sim \mathcal{D}} [\ell(x, y, h)]$, $\widehat{R}_S(h) = \frac{1}{n} \sum_{i=1}^{n} \ell(x_i, y_i, h)$, respectively.

Using MIL-reduction scheme, we obtain the following:

**Theorem 16.** *Multi-label learning with perfectionistic loss is MIL-reducible.*

This can be derived by the same argument with multi-label learning except for $p = \infty$ (see Sec.H).

The empirical Rademacher complexity is bounded as:

**Corollary 17.** *We assume that $\|x_i\| \leq C_2$ for any $i \in [n]$. In the reduced MIL problem, the empirical Rademacher complexity of $\widehat{\mathcal{H}}'$ is given as follows:*

$$\mathfrak{R}_{S'}(\widehat{\mathcal{H}}') = O\left( \frac{\log\left(2n^2 k\right)(bC_1 C_2 \ln(n))}{\sqrt{n}} \right),$$

*where we assume $\|w'\| \leq C_1$.*

Interestingly, we can have the same generalization risk bound with the standard multi-label learning.

The learning algorithm is derived by the following result.

**Corollary 18.** *The reduced ERM of the MIL from multi-label learning with perfectionistic loss is a convex programming problem.*

This is easily obtained by observing the reduction process shown in Sec.H and using Prpoposition 5.

A naive approach for the multi-label learning with perfectionistic loss is to reduce to multi-class learning. That is, we consider all combinations of the multi-label as multi-classes and solve $2^k$-class learning problem with high computational cost. However, by the above corollary, multi-label learning with perfectionistic loss can be solved efficiently.

### 5.2.2 Top-1 ranking learning

Learning to rank is a fundamental problem, and many applications, such as recommendation systems, exist. We consider the following natural scenario in a recommendation problem; a learner has a set that contains several items, and it wishes to recommend an item to a target user from the set.

**Problem setting:** Let $\mathcal{X} \subseteq \mathbb{R}^d$ be an instance space, and $s : \mathcal{X} \to \mathbb{R}$ be a target scoring function. Set $A$ is a finite set of instances selected from $\mathcal{X}$. The learner receives the sequence of the sets of items and the chosen item $S = (A_1, x_1^*), \ldots, (A_n, x_n^*)$, where each $x_i^* \in A_i$ is the highest-valued item determined by the target function $s$. $k$ denotes the average size of the item sets in $S$, that is, $k = \frac{1}{n} \sum_{i=1}^n |A_i|$. Each sample set of items is drawn i.i.d. from $\mathcal{X}$ according to an unknown distribution $\mathcal{D}$ over $2^{\mathcal{X}}$. Assume that the learner predicts the item from the item set using the hypothesis $h \in \mathcal{H} = \{A \mapsto \arg\max_{x \in A} \langle w, x \rangle \mid w \in \mathbb{R}^d\}$.[7] Let $\ell(A, x^*, h)$ is a convex upper bound on the zero-one loss function $I(y \neq \hat{y})$. Equivalently, we consider the zero-one loss $I(\langle w, x^* \rangle - \max_{x \in A \setminus x^*} \langle w, x \rangle \leq 0)$ and its convex upper bounded loss $\ell : (A, x^*, h) \mapsto \Gamma(\langle w, x^* \rangle - \max_{x \in A \setminus x^*} \langle w, x \rangle)$ where $\Gamma : \mathbb{R} \to [0, 1]$ is a convex, nonincreasing and $a$ Lipschitz function. The goal of the learner is to find $h \in \mathcal{H}$ with a small misranking risk w.r.t. the target $s$. Thus, the generalization and empirical risks are formulated as follows:

$$R_{\mathcal{D}}(h) = \mathop{\mathbb{E}}_{A \sim \mathcal{D}} [\ell(A, x^*, h)], \widehat{R}_S(h) = \frac{1}{n} \sum_{i=1}^n \ell(A, x_i^*, h),$$

where $x^* = \arg\max_{x \in A} s(x)$.

We obtain the following by using MIL-reduction scheme:

**Theorem 19.** *Top-1 ranking learning is MIL-reducible.*

---

[7]We consider an $\arg\max$ with a fixed tie-breaking rule.

The reducible condition is satisfied when we set $\alpha(A, x^*) = (x', y')$ where $x' = \{x - x^* \mid x \in A \setminus x^*\}$ $y_i' = -1$ for all $i \in [n]$. The details of the reduction process is in Sec.I.

The empirical Rademacher complexity bound is as follows:

**Corollary 20.** *We assume that $\|x\| \leq C_2$ for any $x \in A_i, \forall i \in [n]$. In the reduced MIL problem, the empirical Rademacher complexity of $\widehat{\mathcal{H}}'$ is given as follows:*

$$\mathfrak{R}_{S'}(\widehat{\mathcal{H}}') = O\left(\frac{\log\left(\hat{a}^2 n^2 (k-1)\right)\left(\hat{a}\ln(2\hat{a}^2 n)\right)}{\sqrt{n}}\right),$$

*where $\hat{a} = 2aC_1 C_2$ and we assume $\|w'\| \leq C_1$.*

The generalization bound can be derived by applying $r_{S'} = k - 1$ and using the fact that $\|z\| \leq 2C_2$ for any $z \in x_i', \forall i \in [n]$ in the reduced MIL. By using Corollary 2, we can obtain the generalization risk bound for the Top-1 ranking learning.

The learning algorithm is designed by the following result:

**Corollary 21.** *The reduced ERM of MIL from top-1 ranking learning is a convex programming problem.*

The corollary can be easily derived from the reduction process detailed in I.

**Extension:** We consider *top-1 ranking learning with negative feedback* which is an extension of top-1 ranking learning. We show the details in Sec.J. Remarkably, the ERM problem of the reduced MIL is a DC programming problem.

## 6 KERNELIZED EXTENSION

Although we consider a linear function set as $\mathcal{G}$; in practice, a nonlinear kernel is required for various learning tasks. A straightforward method is to employ a kernel-approximation technique [see, e.g., Sec.6.6 in Mohri et al., 2018], which constructs feature vectors $\Phi(x) \in \mathbb{R}^D$ with the theoretical guarantee that $\langle \Phi(x_1), \Phi(x_2) \rangle \approx K(x_1, x_2)$ for a user-determined dimension $D$. However, we can use only a limited number of kernels via the approximation technique. Therefore, we show the kernelized version of the reduction.

### 6.1 SETTINGS

We assume that an original problem is defined by $\mathcal{H}, \ell, \mathcal{X}, \mathcal{Y}$, and $\Phi : \mathcal{X} \to \mathbb{H}$, where $\mathbb{H}$ is a reproducing kernel Hilbert space associated to $K(x_1, x_2) = \langle \Phi(x_1), \Phi(x_2) \rangle$. Aside from the computability, we can virtually consider the sample as $S = ((\Phi(x_1), y_1), \ldots, (\Phi(x_n), y_n))$. The ERM-reducible condition is that there exist $(x', y') = \alpha(\Phi(x), y)$, $h = \beta(h')$ and $\ell'$ that satisfies $\ell(\Phi(x), y, h) = \ell'(x_i', y_i', h')$ for any $(x, y) \in \mathcal{X} \times \mathcal{Y}$.

Let $S' = ((x_1', y_1'), \ldots, (x_n', y_n'))$ and let $\mathcal{G} = \{g : z \mapsto \langle w', z \rangle \mid w' \in \mathbb{H}'\}$. We assume that $(\mathcal{H}, \ell)$ is MIL-reducible to $\mathcal{H}', \ell'$. The ERM of the reduced MIL is formulated as:

$$\min_{w' \in \mathbb{H}'} \lambda \|w'\|_{\mathbb{H}'} + \mathcal{L}_{w'}, \qquad (5)$$

where $\mathcal{L}_{w'} = \sum_i^n f_1 \left( y_i' \Psi_p \left( \{ f_2 \left( \langle w', z \rangle \mid z \in x_i' \right) \} \right) \right)$.

### 6.2 COMPUTABILITY

We show that the representer theorem holds for the optimization problem (5).

**Theorem 22** (Representer theorem). *An optimal solution of the ERM problem (5) has the form* $\tilde{w}' = \sum_{z \in P_{S'}} \mu_z z$, *where* $P_{S'} = \bigcup_{i=1}^n x_i'$.

Thus, the ERM problem (5) is equivalently formulated as:

$$\min_{\boldsymbol{\mu} \in \mathbb{R}^{|P_{S'}|}} \lambda \sum_{z, \hat{z} \in P_{S'}} \mu_z \mu_{\hat{z}} \langle z, \hat{z} \rangle + \mathcal{L}_{\boldsymbol{\mu}},$$

where $\mathcal{L}_{\boldsymbol{\mu}} = \sum_{i=1}^n f_1(y_i \Psi_p(\{ f_2(\sum_{z \in P_{S'}} \mu_z \langle z, \hat{z} \rangle) \mid \hat{z} \in x_i' \}))$.

Therefore, if $\langle z_1, z_2 \rangle$ is polynomial-time computable for any $z_1, z_2 \in x'$ using the original kernel function $K$ as an oracle, the ERM of the MIL can be solved similar to linear case according to the condition in Proposition 5 and 6 (DC algorithm for the kernel version is in Sec. L). For all MIL-reducible problems introduced in the paper, $\langle z_1, z_2 \rangle$ is polynomial-time computable using $K$ (see details in Sec.M). Moreover, we can construct $\beta$ in polynomial time.

## 7 DISCUSSION

### 7.1 RELATED WORK

**Other reduction techniques:** Several machine-learning reduction schemes exist [see, e.g., Beygelzimer et al., 2015], and we found general reduction schemes, such as [Pitt and Warmuth, 1990, Beygelzimer et al., 2005]. A major difference between the proposed scheme and existing approaches is that we focus on the reduction of ERM. Various applications of machine learning reductions, such as reduction from multi-class learning to binary classification [James and Hastie, 1998, Ramaswamy et al., 2014], and from ranking to binary classification [Balcan et al., 2008, Ailon and Mohri, 2010, Agarwal, 2014], exist. To the best of our knowledge, the reduction to MIL has not yet been discussed.

**Multi-Class Learning:** Recently, Lei et al. [2019] achieved $\log(k)$-dependent generalization bound. The proposed generalization bound is competitive with the bound. However, our derivation is highly simpler than the analysis of [Lei

et al., 2019] because the reduction allows us to apply the existing MIL bound of [Sabato and Tishby, 2012].

**Complementarily-labeled learning:** Ishida et al. [2017] provided the generalization risk bound in the case in which the training sample contains only complementarily labeled instances (i.e., $\theta = 0$). The proposed generalization bound is incomparable to the bound (see details in Sec.N). Ishida et al. [2017] selected nonconvex loss functions and optimized the empirical risks using a gradient-based algorithm in practice. However, there is no guarantee of the optimality of the solution. We show that the learning problem can be solved by DC algorithm and guarantee the local optima. Moreover, in the special case that sample contains only complementarily labeled data, the learning problem becomes convex programming and we can obtain global optima. To the best of our knowledge, the provided learning algorithm is a first polynomial-time algorithm in the special case.

**Multi-label learning:** Various approaches and generalization analyses have been provided [Yu et al., 2014, Bhatia et al., 2015, Xu et al., 2016a,b]. However, to the best of our knowledge, this paper is the first to propose a $\log(k)$-dependent generalization bound for the linear (or nonlinear kernel) hypothesis class, where $k$ is the number of classes.

**Multi-task learning:** A similar generalization bound was reported by [Pontil and Maurer, 2013]. Their results suggest the advantage of regularizing the weights $w_1, \ldots, w_T$ over $T$ tasks. However, our result is derived from an entirely different argument from [Pontil and Maurer, 2013] and the derivation is highly simplified.

**Top-1 ranking learning:** Top-1 ranking measure was originally discussed in [Hidasi and Karatzoglou, 2018]. However, the basic problem setting is different from ours. They assumed that the recommender has i.i.d. positive and negative items as the sample. Moreover, they did not propose a general form of the problem and theoretical analysis.

**MIL:** MIL was originally proposed by Dietterich et al. [1997], which is known as weakly supervised learning and there have been proposed many real applications [Gärtner et al., 2002, Andrews et al., 2003, Zhang et al., 2013, Doran and Ray, 2014, Carbonneau et al., 2018]. The generalization bound and learning algorithm have been analyzed from the theoretical perspective [Sabato and Tishby, 2012, Doran, 2015, Suehiro et al., 2020]. There have been several studies on the relationship between MIL with other learning tasks. Zhou and Xu [2007] showed that a classical MIL can be considered as specific semi-supervised learning. Zhang et al. [2020] utilized MIL for extracting causal instances. However, these works do not imply any type of reduction in the sense of computation theory: if problem A is reduced to B, then we should immediately obtain an algorithm for A from any algorithm for B combined with the reduction (input-output transformations) with a certain performance guarantee. Suehiro et al. [2020] found that a local-feature-

based time-series classification problem can be reduced to a MIL problem with a generalization risk bound. However, the reduced problem is too specific. Our results first show that various learning problems can be reduced to MIL.

## 7.2 PRACTICAL IMPLICATIONS

An important contribution of the paper in both the theoretical and practical aspects is to provide a simple and general reduction scheme among various learning problems with theoretical guarantees on generalization bounds. This means that when faced with a new learning problem A, we can search for an existing ERM problem B that is reducible from A. If succeeded, then we immediately obtain a learning algorithm for A with a generalization bound. Usually, this process is expected to be much easier than designing a learning algorithm from scratch.

In particular, we demonstrate that various learning problems are reducible to a particular problem, MIL. That is, we only have to improve ERM algorithms for MIL, which work on the original learning problems as well. Moreover, we show that ERM for MIL can be formulated as DC programming problems in Section 4.3. Therefore, we can employ a state-of-the-art DC programming package, which is rapidly evolving these days [Le Thi and Dinh, 2018]. For instance, complementarily labeled learning, which is only known to have a non-convex optimization formulation [Ishida et al., 2017, 2019], would enjoy the benefits from a promising DC programming approach.

**Experiments:** We demonstrate that our theoretical results are practically useful in the following experiment on complementarily labeled learning tasks [8]. We use three artificial datasets and four benchmark datasets available in UCI machine learning repository [9]. The details of artificial datasets are described in Section O. For all datasets, all training instances are complementarily labeled uniformly at random. That is, the ERM problem which is derived from our MIL-reduction scheme becomes a convex programming problem (quadratic programming problem). On the other hand, [Ishida et al., 2017] solves a nonconvex optimization problem by using Adam [Kingma and Ba, 2014]. The size of training sample is fixed to 1000 and we used the remaining data as a test set. Although we did not tune the optimization hyperparameters of [Ishida et al., 2017] (the number of epochs is 200 and the learning rate is 0.01), we stopped the learning at the epoch when the test accuracy was the maximum. The loss of [Ishida et al., 2017] was fixed to PC loss which was the best-performed loss [see Ishida et al., 2017]. Our regularization parameter is chosen from $\{0.01, 1, 100\}$ and the regularization parameter of [Ishida

Table 1: Average test accuracy over 10 trials.

| Dataset | Class | Dim. | Ours | Ishida+ |
|---------|-------|------|------|---------|
| artificial1 | 5 | 50 | **0.9999** | 0.9998 |
| artificial2 | 10 | 50 | **0.808** | 0.646 |
| artificial3 | 25 | 50 | 0.063 | **0.065** |
| covertype | 7 | 54 | **0.562** | 0.549 |
| satimage | 7 | 36 | **0.804** | 0.751 |
| waveform | 3 | 40 | **0.833** | 0.832 |
| yeast | 10 | 8 | 0.348 | **0.407** |

et al., 2017] is chosen from $\{0.01, 1, 100\}$. We evaluated the average accuracy over 10 trials.

Table 1 shows that our method achieved higher classification accuracy than [Ishida et al., 2017] on many datasets. This result indicates that our MIL-reduction scenario for complementarily labeled learning, which is derived from the proposed MIL-reduction scheme, is useful in practice. Moreover, our ERM algorithm does not require any hyperparameters for the optimization because the optimization problem is a convex programming problem (or DC programming problem when the training sample contains both labeled and complementarily labeled instances). On the other hand, the learning algorithm provided by Ishida et al. [2017] solves a nonconvex optimization problem and usually requires several hyperparameters (e.g., learning rate and the number of epochs) of the nonconvex-optimization solver.

## 7.3 CONCLUSION AND FUTURE WORK

We revealed that various learning problems can be reduced to a MIL problem by our ERM-based reduction scheme. The results imply that our MIL-reduction gives a simplified and unified scheme for the analyses for various learning problems. Moreover, we obtained novel theoretical results for some learning problems. A practical concern is that the applicable loss functions are limited in the current scheme. For example, some loss functions without satisfying the conditions of MIL-reducibility (e.g., square loss) cannot be used. We explore the relaxation of the ERM-reducible condition. An interesting open problem is how the class of MIL-reducible problems is characterized. Our results imply that MIL is one of the hardest problems in a certain class C of learning problems. In other words, we could say that MIL is a C-complete problem. We would like to investigate how the class C is characterized.

## ACKNOWLEDGMENT

This work was supported by JSPS KAKENHI (Grant Number JP19H04067 and JP20H05967) and JST, ACT-X (Grant Number JPMJAX200G).

---

[8]The code is available in `https://github.com/suehiro93/MIL_reduction`

[9]`https://archive.ics.uci.edu/ml/`

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
