# OpenReview forum: "Simplified and Unified Analysis of Various Learning Problems by Reduction to Multiple-Instance Learning"
_auai.org/UAI/2022/Conference — UAI 2022 Poster_

### Official Review · Reviewer_RoJ4 · 2022-04-11

**Q2(1) Originality/Novelty:** 2
**Q2(2) Significance/Impact:** 2
**Q2(3) Correctness/Technical Quality:** 2
**Q2(6) Clarity Of Writing:** 3
**Q6 Overall Score:** 5
**Q8 Confidence In Your Score:** 3

**Q1 Summary And Contributions:**

The paper is a theoretical paper without experiments.   It tries to include several popular machine learning ML problems into a Multiple-Instance Learning (MIL)-reduction framework and provides the corresponding generalization bounds.  The considered ML problems consist of multi-class learning, complementarily labeled learning, multi-label learning with perfectionistic loss, top-1 ranking learning, and top-1 ranking learning with negative feedback.

**Q2 Assessment Of The Paper:**

More detailed information regarding each of these aspects is given below:

**Q2(5) Reproducibility:**

3: Good: Key resources (e.g., proofs, code, data) are available and key details (e.g., proofs, experimental setup) are sufficiently well-described for competent researchers to confidently reproduce the main results.

**Q3 Main Strengths:**

1. A theoretical work tries to reduce several popular machine learning problems into a Multiple-Instance Learning (MIL)-reduction framework.  The corresponding theorems and generalization risk bound are provided.

**Q4 Main Weakness:**

1. The work seems a little outdated.  An essential assumption of the framework is that the data features are given while different loss functions are provided to reduce to the MIL framework.  Though generalization risk bounds are provided accordingly, it does not solve the problems, e.g., the data features may need more transformation.  Though kernelization has to be introduced for the reduction, it omits the recent power of deep learning techniques.
2. Though the paper has provided many theoretical results.  Some theoretical proof lacks sufficient explanation.
3. The paper lacks empirical evaluation to demonstrate how the effectiveness of the proposals.  Even without empirical evaluation on real-world classification performance, it should provide some empirical evaluations related to the generalization risk bounds or more elaborations on the tightness of the provided bounds to the existing theoretical results.

**Q5 Detailed Comments To The Authors:**

Overall, the paper has clearly presented the work by providing various theoretical results.  It can be further improved by the following aspects:

1. It is not sure the significance of Multiple-Instance Learning (MIL)-reduction.  The reviewer expects to see how MIL-reduction can indeed improve the performance of the considered machine learning tasks.
2. The paper has provided many theoretical results.  However, many theorems lack detailed proofs while some proofs are not clear for the derivation.  For example,
    2.1. In Theorem 7, the justification of f_1 and f_2 is not clear.  Moreover, how to utilize \Gamma in the proof in pp. 4 is not clear.
    2.2 The reviewer is not sure whether the theoretical results are copied from existing work or all are novelly derived.  It is better to justify them and provide solid elaborations.
3. The paper does not provide an empirical evaluation.  It is better to provide some empirical evaluation on how the MIL-reduction helps the performance in real-world applications.  Even without empirical evaluation of real-world performance, it is interesting to know the tightness of the generalization risk bounds.

Minor comments:

1. DC programming: It is better to clarify the full name of DC programming when it appears for the first time.

**Q7 Justification For Your Score:**

Overall, the paper has clearly presented a framework to include several standard machine learning problems into a Multiple-Instance Learning (MIL)-reduction framework.  Though some theorems and generalization risk bounds are provided, the explanation of the theoretical contributions is not clearly articulated and there is no empirical evaluation to justify the real-world performance of the proposal.

**Q9 Complying With Reviewing Instructions:**

1: Yes.

---

### Official Review · Reviewer_QzBE · 2022-04-13

**Q2(1) Originality/Novelty:** 2
**Q2(2) Significance/Impact:** 2
**Q2(3) Correctness/Technical Quality:** 3
**Q2(6) Clarity Of Writing:** 3
**Q6 Overall Score:** 5
**Q8 Confidence In Your Score:** 2

**Q1 Summary And Contributions:**

This theoretical paper reduces several learning problems to the uniform formulation of multiple-instance learning (MIL). Based on this reduction, a unified and simplified framework is built for analyzing and comparing these problems. The theoretical results are solid, and the paper is well-written.

**Q2 Assessment Of The Paper:**

More detailed information regarding each of these aspects is given below:

**Q2(4) Quality Of Experiments (Optional):**

1: Poor: The experimental evaluation is flawed or the results fail to adequately support the main claims.

**Q2(5) Reproducibility:**

3: Good: Key resources (e.g., proofs, code, data) are available and key details (e.g., proofs, experimental setup) are sufficiently well-described for competent researchers to confidently reproduce the main results.

**Q3 Main Strengths:**

(a). This paper proposes a unified and simplified reduction framework that could use the formulation of MIL to represent several mainstream learning problems, including multi-class learning, complementarily labeled learning, multi-label learning, and top-1 rank learning.
(b). The definitions of the proposed method are clear. The problem formulations of MIL, MIL-reducibility, and the Rademacher complexity bound are presented in detail, making the paper easier to understand.
(c). The theoretical results are promising. Through the proposed reduction, some new insights of generalization bound are presented.

**Q4 Main Weakness:**

(a). This work contains purely theoretical results. The proposed reduction framework does not inspire new algorithms. And the practical applicability of the proposed MIL reduction framework is not clear.
(b). The disadvantages and limitations are not discussed. For example, does this reduction framework bring some drawbacks to existing learning problems?

**Q5 Detailed Comments To The Authors:**

Some practical algorithms should be explored more. Some experiments to verify the advantages of this reduction are needed, and limitations should be discussed.

**Q7 Justification For Your Score:**

This paper proposes a unified reduction framework for some learning problems, which may inspire more studies. However, some practical algorithms are missing.


**Q9 Complying With Reviewing Instructions:**

1: Yes.

---

### Official Review · Reviewer_rdRj · 2022-04-25

**Q2(1) Originality/Novelty:** 4
**Q2(2) Significance/Impact:** 3
**Q2(3) Correctness/Technical Quality:** 3
**Q2(6) Clarity Of Writing:** 2
**Q6 Overall Score:** 6
**Q8 Confidence In Your Score:** 4

**Q1 Summary And Contributions:**

This paper studies the theoretical properties of Multi-Instance Learning (MIL). Built upon results of Sabato and Tishby (2012), this work discusses how multi-class classificaiton, complementary label learning, multi-label learning are reducible to MIL. As the reductions are Rademacher complexity preserving, novel generalization bounds are offered based on theoretical results from the reduced MIL problem.

**Q10 Ethical Concerns (Optional):**

Not applicable.

**Q2 Assessment Of The Paper:**

More detailed information regarding each of these aspects is given below:

**Q2(5) Reproducibility:**

3: Good: Key resources (e.g., proofs, code, data) are available and key details (e.g., proofs, experimental setup) are sufficiently well-described for competent researchers to confidently reproduce the main results.

**Q3 Main Strengths:**

1. This work presents novel theoretical results.
2. The theoretical results link MIL with several classical learning problems, which may be useful for researchers of broader interests.


**Q4 Main Weakness:**

The writing and readibility of this paper could use some improvement. See detailed comments for more.

Some work that links MIL to semi-supervised learning, potential outcome framework are missing in the related work section.

It does not provide discussions on the practical implications of the new theoretical results.

**Q5 Detailed Comments To The Authors:**

1. It is better to clarify whether the results in Section 3, i.e., Proposition 1, are new. If not, it may be moved to Section 2 Preliminaries.

2. Section 4.2 could be merged with Section 4.1.

3. Theorem 3 (application of Sabato and Tishby 2012) can be more specific to improve readability. It seems to be the application of Theorem 20 of Sabato and Tishby.

4. In Section 4.4, "For the reduced MIL problems that satisfy some conditions, we can immediately design a learning algorithm according to the condition." This sentence is rather uninformative.

5. When discussing the reductions, for example multi-class learning to MIL in Section 5.1.1, the space used by introducing multi-class formulation could be better utlized if a more intuitive discussion of the reduction is provided. This also applies to the multi-label reduction; however, the complimentary label learning problem could use a more detailed introduction as it is relatively new.

6. The term “MIL reduction framework", which is frequently referred to in Section 5, is not clearly defined or discussed in Section 4. In Section 4, MIL is formulized, MIL-reducibility is defined, and theorems are given when the problem is MIL reducible. But in order to be a framework, there should also be discussion regarding when the problem is reducible and the general procedure of constructing the reduction.

7. The impact of this paper could be improved with new discussions on how the reduction results could benefit practical algorithm design. There are some previous efforts in linking unifying MIL with other learning schemes. For example, (a) which links MIL to semi-supervised learning and (b) which links MIL to the potential outcome framework in causal effect estimation, etc.
 (a) Zhou and Xu. On the Relation Between Multi-Instance Learning and Semi-Supervised Learning. ICML 2008.
 (b) Zhang, Liu, and Li. Robust Multi-Instance Learning with Stable Instances. ECAI 2020.




**Q7 Justification For Your Score:**

This paper presents novel theoretical results in MIL. It's main drawbacks are in the writing clarity. Furthermore, it's score can also be improved if there were more discussion/results on the empirical significances of the new theoretical results. For example, implications for designing algorithms to solve multi-instance, multi-class, multi-label, complementary label problems.

**Q9 Complying With Reviewing Instructions:**

1: Yes.

---

### Decision · Program_Chairs · 2022-05-15

**Decision:**

Accept (Poster)

**Comment:**

Meta Review: The authors have made detailed responses and clarified several concerns raised in the authors' responses. Novel theoretical results which link multiple-instance learning (MIL) with other well-established learning frameworks are presented in this paper. Correspondingly, some new insights on the generalization risk bound are provided.

There have been some studies on the relationship between MIL with other learning frameworks (e.g. semi-supervised learning), which should be further discussed in this paper.